# Effect of Purple Sweet Potato Using Different Cooking Methods on Cytoprotection against Ethanol-Induced Oxidative Damage through Nrf2 Activation in HepG2 Cells

**DOI:** 10.3390/antiox12081650

**Published:** 2023-08-21

**Authors:** Dagyeong Kim, Yoonjeong Kim, Younghwa Kim

**Affiliations:** Department of Food Science and Biotechnology, Kyungsung University, Busan 48434, Republic of Korea; ghdrlekrud@hanmail.net (D.K.); ang1569@naver.com (Y.K.)

**Keywords:** purple sweet potato, cooking method, hepatoprotection, alcohol damage, Nrf2 pathway

## Abstract

The aim of this study was to investigate the effects of different cooking methods on the hepatoprotective effects of purple sweet potatoes against alcohol-induced damage in HepG2 cells. Purple sweet potatoes (*Ipomeoea batatas* L. Danjami) were subjected to different cooking methods, including steaming, roasting, and microwaving. Steaming resulted in a higher cytoprotective effect against alcohol damage than the other cooking methods. Additionally, the highest inhibition of glutathione depletion and production of reactive oxygen species against alcohol-induced stress were observed in raw and/or steamed purple sweet potatoes. Compared to roasted and/or microwaved samples, steamed samples significantly increased the expression of NADPH quinone oxidoreductase 1, heme oxygenase 1, and gamma glutamate-cysteine ligase in alcohol-stimulated HepG2 cells via the activation of nuclear factor erythroid 2-related factor 2. Moreover, ten anthocyanins were detected in the raw samples, whereas five, two, and two anthocyanins were found in the steamed, roasted, and microwaved samples, respectively. Taken together, steaming purple sweet potatoes could be an effective cooking method to protect hepatocytes against alcohol consumption. These results provide useful information for improving the bioactive properties of purple sweet potatoes using different cooking methods.

## 1. Introduction

Alcohol over-consumption can lead to severe alcoholic liver disease (ALD) [1]. Ethanol, the main component of alcoholic beverages, is metabolized by alcohol dehydrogenase and catabolized to acetaldehyde by cytochrome P450 2E1 (CYP2E1) [2]. CYP2E1 is associated with the generation of reactive oxygen species (ROS) during alcohol metabolism [3]. The enhancement of CYP2E1 causes oxidative stress and cell death via glutathione (GSH) depletion [4]. Previous studies have suggested that antioxidants can prevent liver injury through the modulation of the endogenous antioxidant enzyme system and the inhibition of CYP2E1 expression [5]. Moreover, ethanol consumption decreases the expression or activation of antioxidants and enzymes, including catalase (CAT), glutathione (GSH), glutathione peroxidase (GPx), superoxide dismutase (SOD), and glutathione reductase (GR) [6]. Therefore, antioxidants may be an effective strategy for preventing ALD via the induction of antioxidant enzymes.

The overproduction of ROS can result in oxidative damage to biomolecules, such as DNA, proteins, and lipids, leading to cell death [7,8]. Generally, the activation of nuclear factor erythroid 2-related factor 2 (Nrf2) is considered protective against alcohol toxicity. Nrf2 is a transcription factor, which binds to the antioxidant response element (ARE) and, thereby, modulates the expression of antioxidant enzymes, such as gamma glutamate-cysteine ligase (GCLC), NAD(P)H quinone oxidoreductase 1 (NQO1), and heme oxygenase-1 (HO-1), involved in cellular antioxidant [9,10]. Moreover, various studies have shown that Nrf2 activation increases cell viability against ethanol damage and attenuates ALD [11,12]. Therefore, the Nrf2/ARE pathway is considered a key endogenous antioxidant signaling pathway against alcohol-induced damage.

Cooking methods cause changes in chemical composition, texture, and physical properties [13,14]. Sweet potatoes are usually cooked using different cooking methods before consumption according to culinary traditions worldwide. Several cooking methods for mature stored potato tubers have led to decreased polyphenol content and antioxidant activity [15]. In addition, blanched vegetables have significantly reduced antioxidant activity [16]. Therefore, cooking sweet potatoes using different methods can affect the retention of bioactive components and properties.

Sweet potatoes (*Ipomoea batatas* L.) originate in Central and South America and are cultivated mainly in warm temperate regions. They are cultivated in more than 100 countries and are regarded as an important crop worldwide [17]. Sweet potatoes are an excellent source of starch, fiber, minerals, vitamins, and other nutrients. Purple sweet potato contains abundant anthocyanins and is increasingly favored by consumers because of its health benefits, including antioxidant, antitumor, and hypoglycemic properties [18,19,20]. Im et al. reported that purple sweet potato contains a high content of anthocyanins, such as peonidin and cyanidin [18]. Although previous studies have suggested the antioxidant and cytoprotective effects of purple sweet potato, further studies are required to clarify the hepatoprotective mechanism of purple sweet potato with various cooking methods. The purpose of this study was to investigate the effects of purple sweet potato using different cooking methods (steaming, microwaving, and roasting) on hepatoprotection via the Nrf2 pathway in hepatocytes.

## 2. Materials and Methods

### 2.1. Plant Materials and Chemicals

Purple sweet potatoes (cultivar: *Ipomoea batatas* L. Danjami) were purchased from a local farm (Haenam, Jeonnam, Republic of Korea) in March 2020. 2′,7′-Dichlorofluorescein diacetate (DCF-DA) and 3-(4,5-dimethyl-thiazol-2-yl)-2,5-diphenyl-tetrazolium bromide (MTT) were purchased from Sigma-Aldrich (St. Louis, MO, USA). High-performance liquid-chromatography-grade acetonitrile (ACN), isopropyl alcohol (IPA), water, methanol, and ethanol were obtained from Honeywell Burdic & Jackson (Muskegon, MI, USA). Fetal bovine serum (FBS), Dulbecco’s modified Eagle’s medium (DMEM), and penicillin–streptomycin were obtained from Thermo Scientific (Waltham, MA, USA). All other reagents and solvents were of analytical grade.

### 2.2. Cooking Methods

Fresh purple sweet potatoes were washed and cut into 20 × 20 × 20 mm pieces. For each cooking method, 200 g of purple sweet potato cubes was used. Purple sweet potato cubes were steamed for 15 min. A microwave oven without water was used for microwaving. The samples were placed on glass plates and cooked at 2450 MHz and 700 W for 10 min. The samples were roasted in an oven without oil for 10 min. All cooked and raw samples were lyophilized in a LP-20 freeze dryer (Ilshin-Bio-Base, Donducheon, Korea) to achieve the moisture contents of sample to less than 2%. After freeze-drying, the water contents of raw, steamed, roasted, and microwaved sweet potato were 1.37%, 0.65%, 0.67%, and 1.42%, respectively.

### 2.3. Sample Preparation for Cell-Based Assay

To prepare the extracts, lyophilized samples (10 g) were mixed with 500 mL of methanol for 18 h using a shaker at 23 °C. The solutions were passed through Whatman No. 2 filter paper (Whatman International, Kent, UK), and the methanol was removed using a vacuum evaporator (Eyela, Tokyo, Japan) at 38 °C. Each residue was dissolved in dimethyl sulfoxide (DMSO) at a concentration of 100 mg/mL and stored at −80 °C until use. The extraction yield was 13.55% (*w*/*w*) for raw sample, 11.79% for the steamed, 17.20% for the microwaved, and 17.48% for roasted. Samples were diluted with serum-free medium containing 4% ethanol before treatment to cells. The final concentration of DMSO was less than 0.1% (*v*/*v*) in all treatment groups.

### 2.4. LC-QTOF/MS Analysis

Samples (0.1 g) were weighed into a 2.0 mL centrifuge tube and mixed with 1 mL of ACN/IPA/water (3:3:2, *v*/*v*/*v*). The mixture was vortexed for 5 min and sonicated for 1 h at 4 °C. After sonication, the mixture was centrifuged at 10,000× *g* for 10 min at 4 °C, and the mixture was filtered using a 0.22 µm polytetrafluoroethylene filter (Whatman International). All samples were prepared in quintuplicate.

The ultra-high-performance liquid chromatography coupled with quadrupole time-of-flight mass spectrometry (UHPLC-Q-TOF/MS) system consisted of an Agilent 1260 Infinity Ⅱ series (Agilent Technologies, Santa Clara, CA, USA) with a photodiode array detector (PDA) and an Agilent 6530 Q-TOF mass spectrometer (MS) (Agilent Technologies) equipped with an electrospray ionization (ESI) source. Chromatographic separation was conducted on a YMC-Pack Pro C_18_ (4.6 mm × 150 mm, S-3 µm, YMC, Kyoto, Japan). The column oven temperature was 35 °C, and the injection volume was 10 μL. The mobile phase consisted of 0.1% formic acid in water (A) and 0.1% formic acid in ACN (B) with a gradient elution: 0–30 min, 5% B; 30–30.10 min, 35% B; 30.10–32.10 min, 100% B; 32.10–40 min, 5%, at a flow rate of 0.5 mL/min. MS analysis was performed in both positive-ion and negative-ion modes using the MS and auto-MS/MS scan modes. The MS parameters were as follows: mass range, 10–1700 *m*/*z*; collision energy, 0, 10, and 20 V; drying gas, 10 L/min; gas temperature, 300 °C; sheath gas temperature, 300 °C; fragmentor, 200 V; nebulizer, 45 psi; 12 L/min; capillary, sheath gas flow, 4000 V; and octopole RF Vpp, 750 V.

The data acquisition analysis and data processing were processed using Agilent MassHunter Qualitative Analysis software (Version 10.0, Agilent Technologies). Raw MS data files acquired from UHPLC-Q-TOF/MS analysis were identified via METLIN database B 08.00 and in-house library data based on previous studies [21,22].

### 2.5. Cytotoxicity

HepG2 cells (Korean Collection for Type Cultures, Daejeon, Korea) were cultured in DMEM supplemented with 10% heat-inactivated FBS, 100 U/mL penicillin, and 100 μg/mL streptomycin in a humidified 5% CO_2_ incubator at 37 °C based on previously reported protocols [23]. The cytotoxicity of the samples was assessed using the 3-(4,5-dimethyl-thiazol-2-yl)-2,5-diphenyl tetrazolium bromide (MTT) assay. The cells were seeded in 96-well plates at 1 × 10^4^ cells per well. The next day, HepG2 cells were treated with ethanol (from 1% to 4%) and/or different concentrations of sample (25, 50, and 100 μg/mL) for 24 h. After treatment, the MTT reagent was treated in each well, and the formazan intensity was assessed spectrophotometrically at 550 nm (Thermo Scientific Ltd., Lafayette, CO, USA).

### 2.6. Measurement of Intracellular ROS

To quantify intracellular ROS, 2′,7′ a dichlorofluorescin diacetate (DCF-DA) fluorescent probe was used as previously described [24]. Briefly, HepG2 cells were seeded in 96-well black plates at 5 × 10^4^ cells per well and exposed to the sample (100 μg/mL) next day. DCF-DA (25 μM) in serum-free medium was treated in the wells for 1 h. Then, the cells were treated with 4% ethanol, and the fluorescence intensity was measured after 2 h using a fluorescence spectrophotometer (BioTek Instruments Inc., Winooski, VT, USA) at 485 nm (excitation wavelength) and 530 nm (emission wavelength).

### 2.7. Measurement of Glutathione and Antioxidant Enzyme Activities

HepG2 cells were seeded in 6-well plates at a density of 1 × 10^6^ cells per well for measurement of the GSH and antioxidant enzyme activities. The next day, hepatocytes were treated with the sample (100 μg/mL) and 4% ethanol for 24 h. GSH levels [25], superoxide dismutase (SOD) [26], catalase (CAT) [27], glutathione reductase (GR) [28], and glutathione peroxidase (GPx) [29] activities were evaluated based on a previously reported method.

### 2.8. Western Blotting Analysis

HepG2 cells were seeded at a density of 1 × 10^6^ cells/well in 6-well plates. The next day, the cells were exposed to the samples (100 μg/mL) and 4% ethanol for 24 h, rinsed with phosphate-buffered saline, and collected through centrifugation. The protein expressions of NQO1, HO-1, GCLC, and Nrf-2 were evaluated using Western blot analysis, as described previously [23]. Equal amounts of proteins (50 μg) were loaded on a 10% sodium dodecyl sulfate–polyacrylamide gel and transferred onto a nitrocellulose membrane (Hybond-Nb, Amersham Pharmacia, GE Healthcare, Buckinghamshire, UK). The membranes were blocked with tris-buffered saline/Tween 20 (TBST) containing 5% skim milk for 1 h and then incubated with 1:1000 dilutions of antibodies against HO-1 (sc-10789), NQO1 (sc-32793), GCLC (sc-390811), Nrf2 (sc-365949), PCNA (sc-56), and β-actin (sc-47778) from Santa Cruz Biotechnology (Santa Cruz, CA, USA) for 2 h at 23 °C. The membranes were rinsed with TBST and incubated with 1:2000 dilutions of horseradish peroxidase-conjugated secondary antibodies (sc-2004 and sc-2005) for 1 h at 23 °C. ECL™ reagents (GE Healthcare Bio-Sciences AB, Uppsala, Sweden) were used to detect the protein signal. Western blot images were obtained using ImageQuant^TM^ LAS 500 (GE Healthcare Bio-Sciences AB), and the intensity of the protein band was measured in Image J 1.54d software (NIH, Bethesda, MD, USA).

### 2.9. Statistical Analysis

All data are representative of three or more independent experiments. Statistical analysis was conducted using one-way analysis of variance (ANOVA) followed by Duncan’s multiple comparison test using SAS version 9.4 (SAS Institute Inc., Cary, NC, USA). Significance was set at *p* < 0.05.

## 3. Results and Discussion

### 3.1. Cytoprotective Effects of Purple Sweet Potato with Different Cooking Methods against Alcohol Damage in Hepatocytes

The cytotoxicity of raw and cooked purple sweet potatoes was evaluated using an MTT assay. No cytotoxicity was observed in HepG2 cells at any concentration of sweet potato extract (25, 50, and 100 µg/mL) compared to the control (Figure 1a). Treatment with ethanol (4%, *w*/*w*) decreased the viability by approximately 50% (Figure 1b). However, treatment with raw and steamed sweet potato extracts increased cell viability against alcohol-induced damage. However, roasted and microwaved sweet potatoes did not significantly increase cell viability, except at the highest concentration (100 µg/mL). These results suggest that raw and steamed purple sweet potatoes attenuate alcohol-induced liver damage. Black raspberry, a rich source of anthocyanins, such as cyanidin derivatives, ameliorates ALD through antioxidant and apoptotic pathways [30]. Anthocyanins have potent antidiabetic, antioxidant, and anticancer properties [31]. In particular, anthocyanins from purple sweet potato are composed almost entirely of acylated anthocyanins, and acylation enhances the stability of anthocyanins to thermal treatment [32,33]. A previous study showed that anthocyanins from purple sweet potatoes prevented hepatic damage [34]. Moreover, purple sweet potato contains high amounts of flavonoids, including quercetin, myricetin, and kaempferol, and these flavonoids are well known for their antioxidant activity [32,35,36]. Therefore, the cytoprotective effects of purple sweet potato may be associated with antioxidants, such as phenolic compounds and anthocyanins. Raw and steamed purple sweet potato showed more effective therapeutic potential in hepatocyte damage treatment than roasted and microwaved samples.

### 3.2. Antioxidative Effects of Purple Sweet Potato with Different Cooking Methods against Alcohol-Induced Damage

Alcohol consumption induces the production of reactive molecules, such as acetaldehyde and ROS. Moreover, excessive ROS generation induced by alcohol can cause severe liver injury [37]. Therefore, we measured the effects of purple sweet potato with different cooking methods on ROS production. Ethanol treatment (4%) significantly increased ROS production, whereas ROS generation significantly decreased in all purple sweet potato samples (Figure 2). The highest inhibition of ROS generation was observed in raw and steamed purple sweet potatoes. During alcohol metabolism, CYP2E1 causes excessive ROS production and decreased glutathione (GSH) levels in the liver. GSH plays a central role in protecting mammalian cells against oxidative stress. As shown in Figure 3, treatment with ethanol significantly decreased GSH levels by approximately 50%, whereas treatment with raw purple sweet potato extract significantly increased GSH levels. Among the cooked purple sweet potatoes, the steamed sample showed the highest level of GSH compared to the roasted and/or microwaved samples. The cellular antioxidant enzyme system shows the protective property against oxidative attack and modulates the antioxidant response [37]. Alcohol treatment decreased the antioxidant enzyme activities, including GPx, CAT, GR, and SOD; however, raw and steamed purple sweet potatoes showed markedly increased antioxidant enzyme activities compared with roasted and microwaved sweet potatoes (Table 1). Liver damage caused by dimethylnitrosamine, a potent hepatotoxin or carcinogen, induces a decrease in GSH levels, which is attenuated by anthocyanins [34]. Tang et al. reported that boiling and deep steaming purple sweet potato had relatively less effect on the antioxidant capacity and retention of phenolics, whereas the greatest influence was observed in roasted purple sweet potato [38]. Steaming is also the best cooking method for increasing the concentrations of polyphenols and antioxidants in certain vegetables [39]. In contrast, roasting and microwaving of shiitake mushrooms were better than blanching, boiling, and steaming for retaining antioxidant compounds and showed higher radical scavenging capacities [40]. Therefore, it seems that different cooking methods lead to different antioxidant retention and capacities according to the food matrix and components. Collectively, purple sweet potato significantly inhibited ROS production and GSH depletion against ethanol-stimulated damage in HepG2 cells by enhancing antioxidant enzyme activities. In addition, steaming seemed to be the best preparation method for purple sweet potatoes to retain their antioxidant components and capacity.

### 3.3. Effects of Purple Sweet Potato with Different Cooking Methods on the Expressions of Antioxidant Enzymes and the Activation of Nrf2

To investigate the cytoprotective mechanism of purple sweet potatoes cooked using different methods against ethanol-induced oxidative damage, the expression levels of phase II enzymes and phosphorylated Nrf2 were measured. HO-1, NQO1, and GCLC are good indicators of the adaptive response to oxidative stress. Treatment with ethanol decreased the levels of HO-1, NQO1, and GCLC; however, all purple sweet potatoes increased the expression levels of these proteins (Figure 4). Interestingly, steamed purple sweet potato led to an increased expression of these proteins, similar to uncooked purple sweet potato. Various evidence showed that naturally occurring antioxidants increase the protein expression of antioxidant enzymes [23,41,42]. In particular, anthocyanins enhance the glutathione levels by transactivating the catalytic subunit promoter of GCLC and increasing the levels of HO-1 and NQO1 [43,44,45].

Nrf2 is an important transcription factor, which modulates the gene expression of antioxidant enzymes, such as NQO1, HO-1, and GCLC. In this study, the activation of Nrf2 by purple sweet potato in response to alcoholic oxidative damage was evaluated in HepG2 cells. Raw and steamed purple sweet potatoes showed significantly decreased Nrf2 protein expression in the cytoplasm (Figure 5a). In contrast, these samples showed dramatically enhanced nuclear Nrf2 protein expression compared to the roasted or microwaved samples (Figure 5b). Nrf2 translocates from the cytosol to the nucleus by dissociating from Kelch-like ECH-associated protein 1 upon oxidative attack [46]. Several antioxidants induce HO-1 expression via Nrf2 activation in ethanol-damaged hepatocytes [23,47]. In addition, it has been demonstrated that Nrf2 is activated by cyanidin-3-glycoside against oxidative stress and activates the extracellular-regulated kinases/Nrf2 antioxidant signaling [48]. Xu et al. reported that the knockout of Nrf2 could lose the cytoprotective effects of gentiopicroside, a potent antioxidant, on the Nrf2 pathway against oxidative damage in HepG2 cells [49]. Our results show that steamed purple sweet potato markedly induced the upregulation of phase II enzymes and Nrf2 activation in response to alcohol stress, similar to the raw sample. Moreover, these results are consistent with the finding that Nrf2 suppresses oxidative damage by regulating the activity of antioxidant enzymes during stress [49]. Therefore, the activation of the Nrf2 pathway could be proposed as a therapeutic approach to reduce oxidative stress through alcohol consumption.

### 3.4. Identification of Bioactive Compounds in Purple Sweet Potato with Different Cooking Methods

UPLC-QTOF/MS was performed to analyze the anthocyanins and phenolic compounds in purple sweet potatoes. A total of 13 compounds, such as anthocyanins and phenolics, were tentatively identified from uncooked and cooked samples in both ESI+ and ESI− modes (Table 2). Twelve and eight compounds were detected in the raw and steamed purple sweet potatoes, respectively. The raw sample included ten anthocyanins (cyanidin-3-sophoroside-5-glucoside, peonidin-3-sophoroside-5-glucoside, cyanidin 3-p-hydroxybenzoyl sophoroside-5-glucoside, peonidin 3-p-hydroxybenzoyl sophoroside-5-glucoside, cyanidin 3-(6′′′-caffeoyl sophoroside)-5-glucoside, cyanidin 3-dicaffeoyl sophoroside-5-glucoside, cyanidin 3-dicaffeoyl sophoroside-p-hydroxybenzoyl sophoroside-5-glucoside, peonidin 3-(6′′′-caffeoyl sophoroside)-5-glucoside, peonidin 3-dicaffeoyl sophoroside-5-glucoside, peonidin 3-caffeoyl sophoroside-p-hydroxybenzoyl sophoroside-5-glucoside) and two phenolic compounds (3,4-di-O-caffeoylquinic acid and 3-caffeoylquinic acid). Eight compounds were detected in steamed purple sweet potatoes, including five anthocyanins (cyanidin-3-sophoroside-5-glucoside, peonidin-3-sophoroside-5-glucoside, cyanidin 3-dicaffeoyl sophoroside-5-glucoside, peonidin 3-(6′′′-caffeoyl sophoroside)-5-glucoside, and peonidin 3-dicaffeoyl sophoroside-5-glucoside) and three phenolic compounds (7-hydroxycoumarin, 3,4-di-O-caffeoylquinic acid, and 3-caffeoylquinic acid). However, only two anthocyanins (cyanidin-3-sophoroside-5-glucoside and peonidin-3-sophoroside-5-glucoside) were detected in the roasted and microwaved samples. Two phenolic compounds (3,4-di-O-caffeoylquinic acid and 3-caffeoylquinic acid) were also detected in the roasted and microwaved samples. The anthocyanin content tended to decrease during cooking. Ten anthocyanins were detected in the raw samples, whereas five, two, and two anthocyanins were detected in the steamed, roasted, and microwaved samples, respectively (Figure 6). Interestingly, the number of anthocyanins was higher in the steamed samples than in the roasted or microwaved samples. Anthocyanins are the major bioactive compounds in purple sweet potatoes and show strong radical-scavenging capacities [32]. Anthocyanins become unstable due to heating, and higher temperature negatively affects the anthocyanin content [50]. Particularly, the thermal processing temperature above 100 °C caused significant anthocyanin degradation [51]. In addition, roasted purple sweet potato retained a lower content of total polyphenol compounds and antioxidant capacities compared to steamed potato [38]. Our results were consistent with those of previous studies [50]. In this study, various anthocyanins and phenolic compounds were identified in purple sweet potatoes, and thermal cooking led to a decrease in the number of anthocyanins.

## 4. Conclusions

In the present study, purple sweet potato exhibited cytoprotective effects against alcohol-induced oxidative stress in hepatocytes. In particular, steaming increased cell viability, ROS inhibition, GSH depletion, and antioxidant enzyme activities, including GR, SOD, GPx, and CAT, against alcohol damage in hepatocytes. Furthermore, treatment with raw material and steamed purple sweet potato more efficiently activated the Nrf2 pathway and enhanced the protein expression of antioxidant enzymes, such as HO-1, NQO1, and GCLC, compared to roasting and microwaving. Twelve and eight bioactive compounds were tentatively identified in raw and steamed purple sweet potatoes, respectively. However, only four bioactive compounds were detected in the roasted and microwaved samples. Overall, this study suggests that steaming can be the method of choice for maintaining the antioxidant activity of purple sweet potatoes. These findings suggest that purple sweet potato has potential as a functional food ingredient to attenuate ethanol-induced liver damage.

## Figures and Tables

**Figure 1 antioxidants-12-01650-f001:**
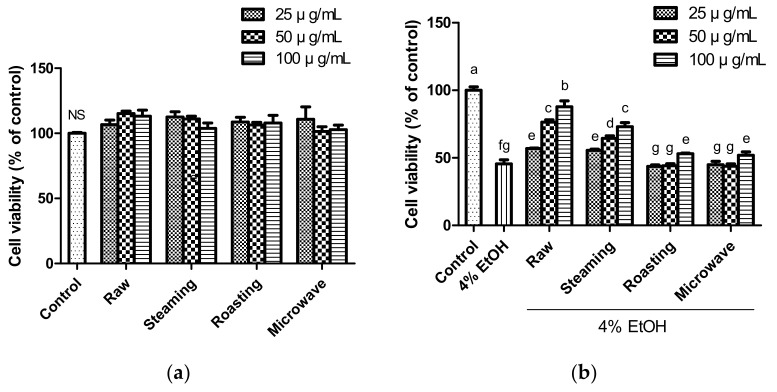
Cytotoxicity of various concentration of purple sweet potato (**a**) and cytoprotective effect (**b**) of various concentration of samples (25, 50, and 100 µg/mL) against 4% ethanol-induced cytotoxicity. The vertical bars represent mean values ± SD and values marked by same letter are not significantly different (*p* < 0.05). NS, not significant.

**Figure 2 antioxidants-12-01650-f002:**
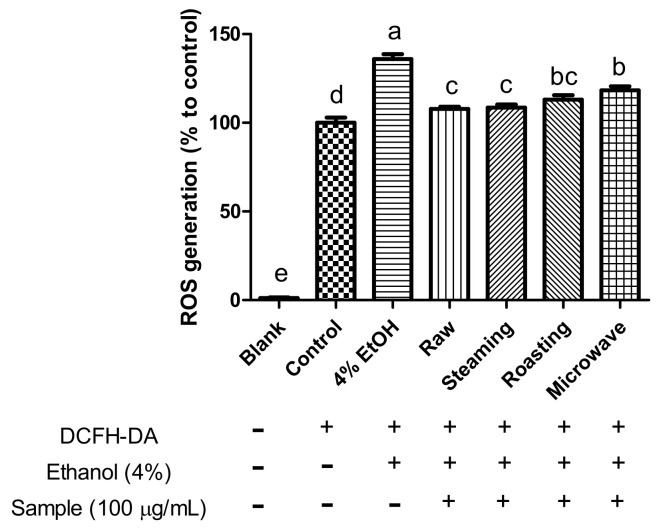
Effect of purple sweet potato with different cooking methods on ethanol-induced ROS in HepG2 cells. The intracellular reactive oxygen species were measured by monitoring fluorescence increase at 120 min. The vertical bars represent mean values ± SD and values marked by same letter are not significantly different (*p* < 0.05).

**Figure 3 antioxidants-12-01650-f003:**
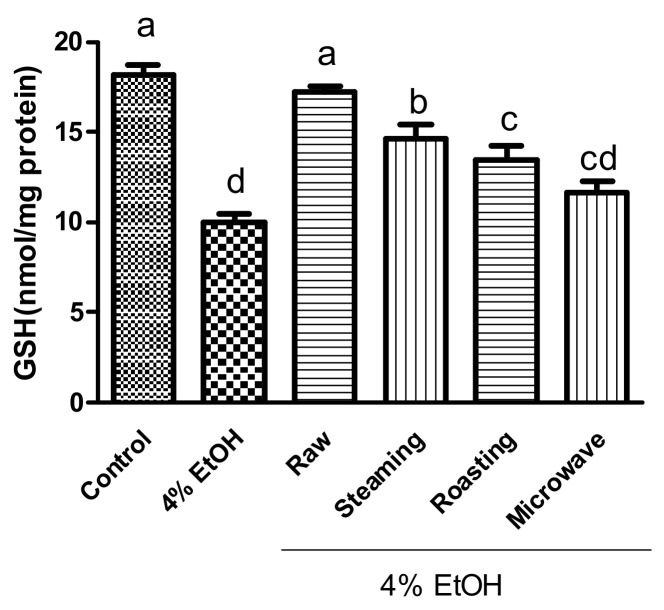
Effects of purple sweet potato with different cooking methods on ethanol-induced GSH depletion in HepG2 cells. The vertical bars represent mean values ± SD and values marked by same letter are not significantly different (*p* < 0.05).

**Figure 4 antioxidants-12-01650-f004:**
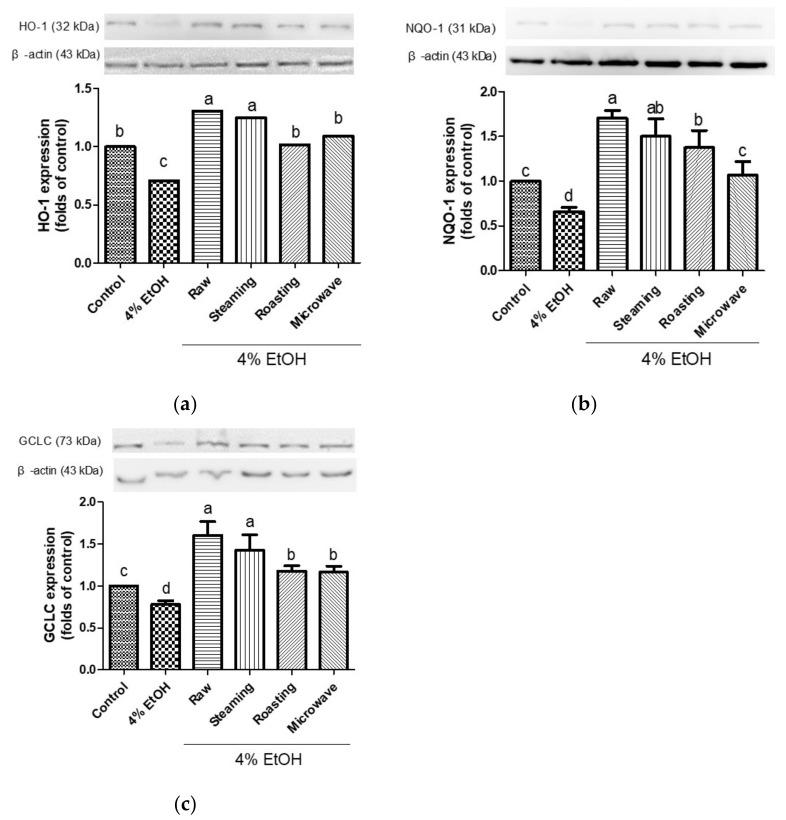
Effects of purple sweet potato with different cooking methods on the protein expression of HO-1 (**a**), NQO1 (**b**), and GCLC (**c**) in ethanol-induced HepG2 cells. β-actin served as internal control. All data are representative of three independent experiments. The vertical bars represent as mean values ± SD and values marked by same letter are not significantly different (*p* < 0.05).

**Figure 5 antioxidants-12-01650-f005:**
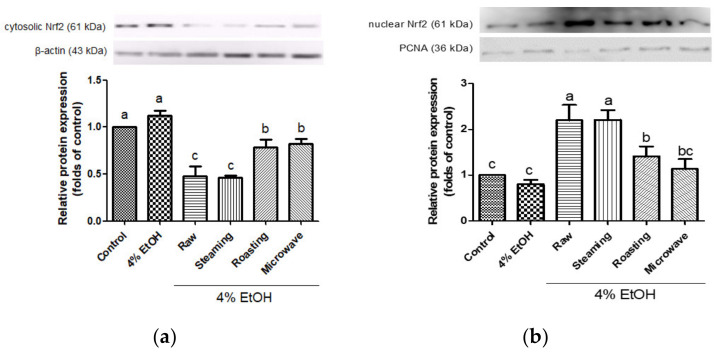
Effects of purple sweet potato with different cooking methods on Nrf2 protein expression in cytoplasm (**a**) and nucleus (**b**). PCNA for nuclear fraction and β-actin for cytoplasmic fraction were used as an internal control. All data are representative of three independent experiments. The vertical bars represent mean values ± SD and values marked by same letter are not significantly different (*p* < 0.05).

**Figure 6 antioxidants-12-01650-f006:**
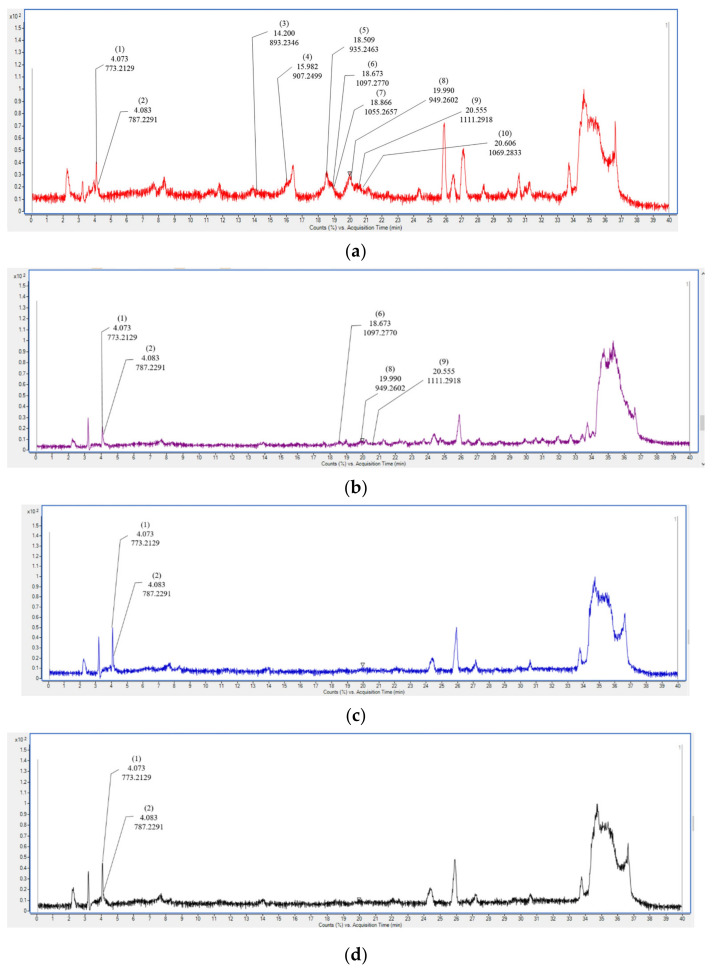
Total ion chromatogram (TIC) of anthocyanins in raw (**a**), steamed (**b**), roasted (**c**), and microwaved (**d**) purple sweet potato by different cooking methods using LC-QTOF/MS analysis. Cyanidin-3-sophoroside-5-glucoside (1), peonidin-3-sophoroside-5-glucoside (2), cyanidin 3-p-hydroxybenzoyl sophoroside-5-glucoside (3), peonidin 3-p-hydroxybenzoyl sophoroside-5-glucoside (4), cyanidin 3-(6′′′-caffeoyl sophoroside)-5-glucoside (5), cyanidin 3-dicaffeoyl sophoroside-5-glucoside (6), cyanidin 3-dicaffeoyl sophoroside-p-hydroxybenzoyl sophoroside-5-glucoside (7), peonidin 3-(6′′′-caffeoyl sophoroside)-5-glucoside (8), peonidin 3-dicaffeoyl sophoroside-5-glucoside (9), peonidin 3-caffeoyl sophoroside-p-hydroxybenzoyl sophoroside-5-glucoside (10).

**Table 1 antioxidants-12-01650-t001:** Effects of purple sweet potato with different cooking methods on antioxidant enzyme activities in HepG2 cells.

Sample		GR ^(1)^	GPx ^(2)^	CAT ^(3)^	SOD ^(4)^
Control		8.132 ± 1.027 ^ab^	25.150 ± 2.911 ^a^	53.718±1.135 ^c^	53.977±1.138 ^bc^
4% Ethanol		4.472 ± 0.422 ^d^	9.847 ± 0.769 ^d^	45.323 ± 2.491 ^d^	25.030 ± 3.758 ^d^
+4% Ethanol	Raw	8.899 ± 0.745 ^a^	21.051 ± 1.818 ^b^	82.618 ± 3.072 ^a^	89.892 ± 14.354 ^a^
	Steaming	7.861 ± 1.075 ^ab^	20.373 ± 0.211 ^bc^	77.386 ± 3.302 ^a^	85.988 ± 14.293 ^a^
	Roasting	6.603 ± 1.011 ^bc^	19.132 ± 1.400 ^bc^	64.606 ± 5.398 ^b^	64.257 ± 9.080 ^b^
	Microwave	5.469 ± 0.451 ^cd^	18.207 ± 0.740 ^c^	68.246 ± 2.732 ^b^	40.143 ± 1.188 ^cd^

^(1)^ Glutathione reductase (GR) (nmol/minute/mg of protein). ^(2)^ Glutathione peroxidase (GPx) (nmol/min/mg protein). ^(3)^ CAT antioxidant enzyme activity (CAT) (µmol/min/mg protein). ^(4)^ Superoxide dismutase (SOD) (units/min/mg of protein). All values represent mean ± SD and values marked by same letter are not significantly different (*p* < 0.05).

**Table 2 antioxidants-12-01650-t002:** Identification of compounds in purple sweet potato with different cooking methods using LC-QTOF/MS.

Compound Class	Ion Mode	Rt (min) ^(1)^	*m*/*z* Calculated ^(2)^	*m*/*z* Observed ^(3)^	Error (ppm)	Fragment Ions (*m*/*z*)	Molecular Formula	Compound Name	Observed Sample ^(4)^
Anthocyanin	POS	4.073	773.21	773.21	−0.77	611/449/287	C_33_H_41_O_21_	cyanidin-3-sophoroside-5-glucoside	R, S, RO, M
		4.083	787.23	787.23	−1.01	625/459/301	C_34_H_43_O_21_	peonidin-3-sophoroside-5-glucoside	R, S, RO, M
		14.20	893.24	893.24	0.11	731/449/287	C_40_H_45_O_23_	cyanidin 3-*P*-hydroxybenzoyl sophoroside-5-glucoside	R
		15.98	907.25	907.25	−0.55	745/463/301	C_41_H_47_O_23_	peonidin 3-*P*-hydroxybenzoyl sophoroside-5-glucoside	R
		18.51	935.25	935.2463	1.18	773/449/287	C_42_H_47_O_24_	cyanidin 3-(6′′′-caffeoyl sophoroside)-5-glucoside	R
		18.67	1097.28	1097.28	−0.78	935/449/287	C_51_H_53_O_27_	cyanidin 3-dicaffeoyl sophoroside-5-glucoside	R, S
		18.87	1055.27	1055.27	1.33	898/449/287	C_49_H_51_O_26_	cyanidin 3-dicaffeoyl sophoroside-p-hydroxybenzoyl sophoroside-5-glucoside	R
		19.99	949.26	949.26	1.07	787/463/301	C_43_H_49_O_24_	peonidin 3-(6′′′-caffeoyl sophoroside)-5-glucoside	R, S
		20.56	1111.29	1111.29	−0.33	949/463/301	C_52_H_55_O_27_	peonidin 3-dicaffeoyl sophoroside-5-glucoside	R, S
		20.61	1069.28	1069.28	2.01	907/463/301	C_50_H_53_O_26_	peonidin 3-caffeoyl sophoroside-p-hydroxybenzoyl sophoroside-5-glucoside	R
Phenolic compound	POS	5.24	162.03	162.03	−0.58	133/117	C_9_H_6_O_3_	7-hydroxycoumarin	S
	NEG	6.30	516.13	516.13	−3.37	353	C_25_H_24_O_12_	3,4-di-*O*-caffeoylquinic acid	R, S, RO, M
		5.23	354.10	354.09	−3.18	191	C_16_H_18_O_9_	3-caffeoylquinic acid	R, S, RO, M

^(1)^ Rt: Retention time in minutes. ^(2)^ *m*/*z* calculated: m/z calculated by software. ^(3)^ *m*/*z* observed: *m*/*z* observed in experiment. ^(4)^ R = Raw purple sweet potato, S = Steamed purple sweet potato, RO = Roasted purple sweet potato, M = Microwaved purple sweet potato

## Data Availability

The data that support the findings of this study are available from the corresponding author upon reasonable request.

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
