# Peer review of "Effect of Purple Sweet Potato Using Different Cooking Methods on Cytoprotection against Ethanol-Induced Oxidative Damage through Nrf2 Activation in HepG2 Cells"

_antioxidants, 2023, doi:10.3390/antiox12081650_

Round 1
Reviewer 1 Report
Dagyeong Kim and Younghwa Kim investigated the effects of different cooking methods on the liver-protective effects of purple sweet potato against alcohol-induced HepG2 cell injury. It was found that steaming produced a stronger cytoprotective effect against purple sweet potato cell injury compared to other cooking methods. This is very interesting. However, there are some problems:
1. In the introduction, the sentence "The enhancement of CYP2E1 causes oxidative stress and cell death by glutathione (GSH) depletion [4]. Previous studies have suggested that an endogenous antioxidant system prevents damage by enhancing CYP2E1 activity [5]. ", they seem contradictory.
2. The Nrf2/ARE pathway that first appeared does not have a full name or introduction.
3. Is the purple sweet potato 20×20×20 mm pieces freeze-dried well, and is the moisture content tested?
4. What is the extraction rate of purple sweet potatoes for each cooking method? (How much extract can be obtained from 1g of raw material respectively?)
5. The gray scale of each of the B-actin bands in Figure 4C and Figure 5B is inconsistent, and replacement is recommended.
6. The WB bands in Figures 5a and 5b are the opposite of what is described in the article (nuclear and cytosol).
7. It is generally believed that microwave heating temperature is not high, heating time is short, and there is no contact with water, less composition loss. The results of reference [15] are the same, but your results are different. Is there a reasonable explanation?
Author Response
Dagyeong Kim and Younghwa Kim investigated the effects of different cooking methods on the liver-protective effects of purple sweet potato against alcohol-induced HepG2 cell injury. It was found that steaming produced a stronger cytoprotective effect against purple sweet potato cell injury compared to other cooking methods. This is very interesting. However, there are some problems:
- In the introduction, the sentence "The enhancement of CYP2E1 causes oxidative stress and cell death by glutathione (GSH) depletion [4]. Previous studies have suggested that an endogenous antioxidant system prevents damage by enhancing CYP2E1 activity [5]. ", they seem contradictory.
--> Answer: We agree with your comment. So, we revised the sentence to “Previous studies have suggested that antioxidants can prevent liver injury through the modulation of endogenous antioxidant enzyme system and the inhibition of CYP2E1 expression [5]. “ (page 1, line 22-24).
- The Nrf2/ARE pathway that first appeared does not have a full name or introduction.
--> Answer: We added the full name of Nrf2 and ARE when it first appeared in introduction (page 1-2, line 40-45).
- Is the purple sweet potato 20×20×20 mm pieces freeze-dried well, and is the moisture content tested?
--> Answer: Raw and cooked sweet potatoes were fully lyophilized for 4 days. Freeze dryer machine we used finishes when the moisture content of food reaches less than 3%. Also, typical food freeze-drying cycles usually target residual moisture contents in the range of 1% to 3% water by weight.
After freeze-drying, water contents of raw, steamed, roasted and microwaved sweet potatoes were 1.37%, 0.65%, 0.67%, and 1.42%, respectively. And we added the freeze dryer information in the manuscript (page 2, line 85-86)..
- What is the extraction rate of purple sweet potatoes for each cooking method? (How much extract can be obtained from 1g of raw material respectively?)
--> Answer: We added the extraction yield of each sample in the manuscript (page 2, line 92-94).
“The extraction yield was 13.55% (w/w) for raw sample, 11.79% for the steamed, 17.20% for the microwaved, and 17.48% for roasted.”
- The gray scale of each of the B-actin bands in Figure 4C and Figure 5B is inconsistent, and replacement is recommended.
--> Answer: We replaced the western blot band in Fig. 4C and 5B according to your comment (page 7, line 264; page 8, line 190)
- The WB bands in Figures 5a and 5b are the opposite of what is described in the article (nuclear and cytosol).
--> Answer: Thank you for your comments. It was our mistake and we corrected (page 8, line 292-293).
- It is generally believed that microwave heating temperature is not high, heating time is short, and there is no contact with water, less composition loss. The results of reference [15] are the same, but your results are different. Is there a reasonable explanation?
--> Answer: We checked the cooking method and found the mistake for microwave cooking time. We corrected the cooking time of microwave (5 min to 10 min) (page 2, line 85).
We think the cytoprotective effects of purple sweet potato may be associated with its antioxi-dants, such as phenolic compounds and anthocyanins. Several studies reported showed that steaming is more effective to possess antioxidant properties [Ref. 38-39]. However, roasting and microwaving of shiitake mushrooms showed higher radical scavenging capacities compared to other cooking methods [40]. Therefore, we think that different cooking methods lead to different antioxidant retention and capacities according to the food matrix and components (page 5, line 219-227).
- Tang, Y.; Cai, W.; Xu, B. Profiles of phenolics, carotenoids and antioxidative capacities of thermal processed white, yel-low, orange and purple sweet potatoes grown in Guilin, China. Food Sci. Hum. Wellness 2015, 4, 123-132, doi:10.1016/j.fshw.2015.07.003.
- Dolinsky, M.; Agostinho, C.; Ribeiro, D.; Rocha, G.D.S.; Barroso, S.G.; Ferreira, D.; Polinati, R.; Ciarelli, G.; Fialho, E. Ef-fect of different cooking methods on the polyphenol concentration and antioxidant capacity of selected vegetables. J. Cu-lin. Sci. Technol. 2015, 14, 1-12, doi:10.1080/15428052.2015.1058203.
- Lee, K.; Lee, H.; Choi, Y.; Kim, Y.; Jeong, H.S.; Lee, J. Effect of different cooking methods on the true retention of vita-mins, minerals, and bioactive compounds in Shiitake mushrooms (Lentinula edodes). Food Sci. Technol. Res. 2019, 25, 115-122, doi:10.3136/fstr.25.115.

Reviewer 2 Report
Review of the manuscript entitled: Effect of Purple Sweet Potato with Different Cooking Methods on Cytoprotection against Ethanol-Induced Oxidative Damage through Nrf2 Activation in HepG2 cells. The manuscript deals with a very important topic, but some corrections are needed. The manuscript is quite neatly done, I do not find any serious errors.
1. In abstract and introduction clear aim of the manuscript should be added e.g. "The aim of the present study was to ...". In case of introduction aim should be at the end of introduction.
2. Lines 68-72 “2.1. Chemicals” - it is unbelievable that only one reagent was needed. Where is the cell culture medium, FBS, PBS, other needed reagents and kits? Key are antibody catalog numbers and their dilutions. Catalog numbers are also needed for key reagents, without these data it is impossible to repeat the experiments. Moreover, for the work to be accurate, data on the cultivation of the plant should be given.
Author Response
Review of the manuscript entitled: Effect of Purple Sweet Potato with Different Cooking Methods on Cytoprotection against Ethanol-Induced Oxidative Damage through Nrf2 Activation in HepG2 cells. The manuscript deals with a very important topic, but some corrections are needed. The manuscript is quite neatly done, I do not find any serious errors.
--> We really appreciate the comments and have learned a lot. We revised the manuscript according to your comments and marked with red color in revised manuscript. Thank you.
- In abstract and introduction clear aim of the manuscript should be added e.g. "The aim of the present study was to ...". In case of introduction aim should be at the end of introduction.
--> Answer: Thank you for your comments. We revised the sentence to clarify the aim of this study in the abstract (page 1, line 9). Also, the purpose of this study is present at the end of introduction (page 2, line 66-68)
- Lines 68-72 “2.1. Chemicals” - it is unbelievable that only one reagent was needed. Where is the cell culture medium, FBS, PBS, other needed reagents and kits? Key are antibody catalog numbers and their dilutions. Catalog numbers are also needed for key reagents, without these data it is impossible to repeat the experiments. Moreover, for the work to be accurate, data on the cultivation of the plant should be given.
--> Answer: We added the detail information of chemicals, cell culture medium, FBS, and other reagents in “materials and methods” part (page 2, line 72-78; page 3, line 122-124). Also, we inserted the information of antibodies including catalog numbers, manufacturer, and dilutions (page 4, line 154-160). Cultivar information of sweet potato (Ipomoea batatas L. Danjami) is present in line 71 (page 2).

Round 2
Reviewer 1 Report
The author explained and revised all the questions I raised. I'm basically satisfied with that. However, regarding question 3, It is difficult of internal drying for 20×20×20 mm pieces (such a large volume).
Author Response
Reviewer’s comment
The author explained and revised all the questions I raised. I'm basically satisfied with that. However, regarding question 3, It is difficult of internal drying for 20×20×20 mm pieces (such a large volume).
A: Thank you very much for your comment. We determined the size of the sweet potato by referring to the domestic cooking methods and previous studies (Ref. 1 & 2). Several previous studies performed similarly to our sample preparation (sweet potato size: 1.5 ~ 2.5 cubic cm). Also, we added information of water contents of sample in the manuscript (page 2, line 87-89) and highlighted. Thank you.
Ref 1. Bellail, A.A. et al., Effect of Home-Cooking Methods on Phenolic Composition and Antioxidant Activity of Sweetpotato (Ipomoea batatas (L.) Lam.) Cultivars Grown in Egypt. Food and Nutrition Sciences, 2012, 3, 490-499
Ref 2. Im, Y.R. et al., Phenolic Composition and Antioxidant Activity of Purple Sweet Potato (Ipomoea batatas (L.) Lam.): Varietal Comparisons and Physical Distribution. Antioxidants 2021, 10, 462.
Ref 3. Amagloh, F.C. et al., Household Processing Methods and Their Impact on Bioactive Compounds and Antioxidant Activities of Sweetpotato Genotypes of Varying Storage Root Flesh Colours. Antioxidants, 2022, 11, 1867
